# Copper Chaperone for Superoxide Dismutase *FoCCS1* in *Frankliniella occidentalis* May Be Associated with Feeding Adaptation after Host Shifting

**DOI:** 10.3390/insects13090782

**Published:** 2022-08-29

**Authors:** Tao Zhang, Li Liu, Jun-Rui Zhi, Yu-Lian Jia, Wen-Bo Yue, Guang Zeng, Ding-Yin Li

**Affiliations:** 1Institute of Entomology, Guizhou University, Guiyang 550025, China; 2Guizhou Provincial Key Laboratory for Agricultural Pest Management in the Mountainous Region, Guiyang 550025, China

**Keywords:** *Frankliniella occidentalis*, host shifting, superoxide dismutase, RNA interference, feeding adaptation

## Abstract

**Simple Summary:**

Western flower thrips (*Frankliniella occidentalis*) have a wide range of hosts. Therefore, they can colonize new host plants with each seasonal change. This study examined whether the superoxide dismutase (*SOD*) gene regulates the feeding adaptation of *F. occidentalis* after host shifting. The coding sequences for *CCS1* and *MnSOD2* in *F. occidentalis* were cloned and the corresponding amino acid sequence was predicted, and the mRNA expression levels of these two genes at different developmental stages were determined. Further, the mRNA expression levels of *FoCCS1* and *FoMnSOD2* in second-instar larvae and adult females transferred to kidney bean and broad bean plants for rearing were analyzed. Decreasing the mRNA levels of *FoCCS1* and *SOD* activity by RNA interference significantly reduced the survival rate and fecundity of adult *F. occidentalis* females. These findings provide a reference for analyzing the adaptive mechanism of *F. occidentalis* after host shifting.

**Abstract:**

Western flower thrips (*Frankliniella occidentalis*) pose a serious threat to the global vegetable and flower crop production. The regulatory mechanism for superoxide dismutase (SOD) in the feeding adaptation of *F. occidentalis* after host shifting remains unclear. In this study, the copper chaperone for SOD (*CCS*) and manganese SOD (*MnSOD*) genes in *F. occidentalis* were cloned, and their expression levels at different developmental stages was determined. The mRNA expression of *FoCCS1* and *FoMnSOD2* in *F. occidentalis* second-instar larvae and adult females of F_1_, F_2_, and F_3_ generations was analyzed after shifting the thrips to kidney bean and broad bean plants, respectively. The F_2_ and F_3_ second-instar larvae and F_2_ adult females showed significantly upregulated *FoCCS1* mRNA expression after shifting to kidney bean plants. The F_1_ second-instar larvae and F_2_ adult females showed significantly upregulated *FoCCS1* mRNA expression after shifting to broad bean plants. The RNA interference significantly downregulated the *FoCCS1* mRNA expression levels and adult females showed significantly inhibited SOD activity after shifting to kidney bean and broad bean plants. *F. occidentalis* adult females subjected to RNA interference and released on kidney bean and broad bean leaves for rearing, respectively, significantly reduced the survival rate and fecundity. These findings suggest that *FoCCS1* plays an active role in regulating the feeding adaptation ability of *F. occidentalis* after host shifting.

## 1. Introduction

The western flower thrips, *Frankliniella*
*occidentalis* Pergande (Thysanoptera: Thripidae) is one of the most difficult-to-manage pests of vegetable and flower crops worldwide. *F. occidentalis* directly damages crop production by feeding on the leaves, flowers, and fruits of host plants and laying eggs on the plants; it can also indirectly damage crops by transmitting plant viruses [1]. This was first reported as an invasive thrip species in China in 2000; since then, it has rapidly spread to more than 10 provinces. *F. occidentalis* can reproduce parthenogenetically and sexually and has a short generation cycle [2,3]. The wide host range (>500 plant species), effective hiding ability, and adaptability of *F. occidentalis* greatly contributes to its management difficulty [4]. These biological features also enable *F. occidentalis* to invade and colonize seasonal crops of the current season. For example, kidney and broad beans are the main hosts of *F. occidentalis* [5]. Kidney beans are mainly cultivated in the summer and autumn seasons and are one of the main summer hosts of *F. occidentalis*. In contrast, broad beans are cultivated in winter and spring and are one of the main winter hosts of *F. occidentalis*. The seasonal growth characteristics of kidney beans and broad beans may provide a habitat for the colonization of *F. occidentalis* in different seasons.

It is a common ecological phenomenon for herbivorous insects to transfer to new hosts for feeding and colonization because of factors such as seasonal alterations in crop growth [6,7,8]. When herbivorous insects who colonize and adapt to one host for a long time shift to another host, they must first overcome the new host plant’s defense barrier to successfully colonize it [8,9]. An apoplastic burst of reactive oxygen species (ROS) occurs in the host plant, which is the first barrier against a continued attack by herbivorous insects [10]. ROS accumulation in host plants directly prevents insect feeding damage, and mediates the activation of defense genes and enzymes by regulating their transcription, participates in the interaction of signaling pathways in the plant defense system and establishes corresponding defense mechanisms [10,11,12]. Among all ROS, H_2_O_2_ is a highly stable and freely diffusible core element of plant-induced defense response [11,12]. In plants, H_2_O_2_ directly acts on herbivorous insects or enters the insect cells and changes the redox state, resulting in oxidative damage in the insect midgut [10,13]. For example, *Schizaphis graminum* infestation in wheat for 20 min leads to H_2_O_2_ accumulation, which effectively protects wheat from subsequent feeding damage [14]. An increase in H_2_O_2_ concentration is also an effective defense mechanism used by the peanut crop against cotton bollworm *Helicoverpa armigera* [15].

Superoxide dismutase (SOD) is a key enzyme in the first-line antioxidant defense system and is the only enzyme that directly metabolizes the superoxide class of ROS [16]. SOD catalyzes the disproportionation of ROS to form H_2_O and H_2_O_2_. H_2_O_2_ is subsequently converted to H_2_O by the action of catalase or peroxidase, which ultimately scavenges ROS and protects the organism from oxidative damage [16]. SOD in insects could be categorized into Cu/ZnSOD and MnSOD. Cu/ZnSOD is abundant in the serum and cytoplasm of the insect midgut, fat body, and body wall, whereas MnSOD mainly exists in the mitochondria [17]. Additionally, MnSOD is usually classified and annotated as SOD2. The Cu/ZnSOD family usually comprises five members: cytoplasmic Cu/ZnSOD, extracellular SOD (SOD3), copper chaperone (CCS), related to SOD (RSOD), and SODesque (SODq) [18]. Studies have shown that CCS is essential for the activation of mammalian Cu/ZnSOD [19]. Moreover, CCS is an essential factor for the regulation of copper homeostasis and promotion of all levels of SOD1 activity maturation [20,21,22]. Studies have indicated that insect SOD plays an important role in regulating the elimination of ROS induced by temperature [23], ultraviolet light, and heavy metal stress [24], and actively participates in the regulation of insect feeding adaptation during host shifting [25].

When insects shift to new hosts for feeding, a series of plant ROS-related defense responses are inevitably triggered [11,26,27]. Insect SOD plays an important role in protecting insects from oxidative damage and regulating their adaptability to host plants [13,17,25]. Previous studies have shown that *F. occidentalis* SODs are actively involved in the feeding response after host shifting [25]. However, the specific molecular mechanism underlying the regulation of the feeding adaptation of *F. occidentalis* after host shifting remains unclear. Therefore, in this study, the *CCS* and *MnSOD* gene subfamily of *F. occidentalis* were cloned and their mRNA expression levels in the second-instar larvae and adult females of the F_1_, F_2_, and F_3_ generations after host shifting were analyzed. Using RNAi technology, the effect of the downregulation of the highly expressed SOD genes on the feeding adaptation of *F. occidentalis* after host shifting was also investigated.

## 2. Materials and Methods

### 2.1. Insect Culture and Host Plant Cultivation

*F. occidentalis* were collected from Huaxi District, Guiyang City, China, and were raised in an RXZ-type multistage programming artificial climate box (Ningbo Jiangnan Instrument Factory, Ningbo, China) at the Institute of Entomology, Guizhou University. The breeding environment was maintained a 25 ± 1 °C, 70 ± 5% relative humidity, and 14:10 h light:dark photoperiod. Kidney bean pods were provided as food. *F. occidentalis* reared for more than 70 successive generations were used as the experimental population.

Kidney bean seeds *(Phaseolus vulgaris* L. (Fabales: Fabaceae), variety Jinshulu, were obtained from the Shengnong Seed Company in Xinji City, China. One seed was planted in each artificial climate chamber (environment conditions: 25 ± 1 °C, 65 ± 10% relative humidity, and 14:10 h light:dark photoperiod) in sterilized nutrient soil, and two seeds were planted in each vegetative bowl (11.5 cm diameter × 10.0 cm height). Approximately 16 days after germination, the second trifoliate leaves appeared, and healthy and undamaged kidney bean plants with similar growth characteristics were selected as the experimental hosts.

Broad bean seeds (*Vicia faba* L.), variety Lincan No. 5, were purchased from Kangle County Jinzhong Agricultural Products Industrial Development Co., Ltd., Linxi, China, and cultivated under the same conditions as that of kidney bean plants. The seedlings were grown for 20 days after germination, and healthy and undamaged broad bean plants with 8–10 leaves and approximately 15 cm in height showing similar growth characteristics were selected as the experimental hosts.

### 2.2. Reverse-Transcription PCR and Cloning by Rapid Amplification of the cDNA Ends of the F. occidentalis SOD Gene

Total RNA from the *F. occidentalis* adults from pod-feeding populations was extracted using Eastp^®^ Super Total RNA Extraction Kit (Promega (Beijing) Biotech Co., Ltd., Beijing, China). cDNA was synthesized by reverse transcription using the RevertAid First Strand cDNA Synthesis Kit (Thermo Scientific, Vilnius, Lithuania) as per the manufacturer’s instructions. SOD sequences were screened on the basis of the tested transcriptome data. The online tool primer designing tool (https://www.ncbi.nlm.nih.gov/tools/primer-blast/index.cgi?LINK_LOC=BlastHome, accessed on 14 March 2019) was used to design cDNA cloning primers *FoC**CS1*-F/*FoC**CS1*-R and *FoMnSOD2*-F/*FoMnSOD2*-R (Appendix A). PCR amplification was performed using Taq PCR Master Mix (2×, with blue dye) (Sangon Biotech, Shanghai, China). The PCR product was purified using the SanPrep Column PCR Product Purification Kit (Sangon Biotech), ligated using the pMD18T vector cloning kit (Takara Bio USA, Inc., San Jose, CA, USA), and transformed into competent DH5α cells. The transformed DH5α cells were cultured in Luria broth (LB) without antibiotics at 37 °C for 1.5 h and then streaked on LB agar plates containing 100 μg/mL ampicillin and incubated. Subsequently, the two intermediate gene fragments were obtained by sequencing after colony PCR and detected by electrophoresis. As per the obtained intermediate fragment, 3′-end primers (Appendix A) were designed according to the instructions of SMARTer^®^ RACE 5′/3′ Kit Components (Takara Bio USA, Inc.). The 3′-end cDNA sequence was obtained through PCR amplification, gel-cutting purification, ligation, transformation, and sequencing. Using SeqMan v.7.1.0 software (DNASTAR, Inc., Madison, WI, USA), the amplified intermediate fragment sequence of the target gene and the 3′-end cDNA sequence were spliced to obtain the complete coding sequence (CDS) of the target gene. The full-length verification primers c*FoCCS1*-F/c*FoCCS1**-*R and c*FoMnSOD2*-F/c*FoMnSOD2*-R (Appendix A) were set at both ends of the CDS region for full-length sequence verification by PCR amplification.

### 2.3. Bioinformatics Analysis of the SOD Gene Sequence

The cloned gene sequences were compared and analyzed using NCBI BLAST (https://blast.ncbi.nlm.nih.gov/Blast.cgi, accessed on 29 May 2019). The complete open reading frame (ORF) of the gene sequence was identified using ORF Finder (https://www.ncbi.nlm.nih.gov/orffinder/, accessed on 29 May 2019). The amino acid sequences were predicted using the Translate tool (http://web.expasy.org/translate/, accessed on 29 May 2019), and phosphorylation sites were predicted with KinasePhos (http://kinasephos.mbc.nctu.edu.tw, accessed on 29 May 2019). Using ExPASy–PROSITE (https://prosite.expasy.org/, accessed on 29 May 2019), protein sequences were analyzed as PROSITE signatures. The ExPASy-ProtParam tool (https://web.expasy.org/protparam/, accessed on 29 May 2019) was used to analyze the physicochemical properties of the predicted proteins. NCBI Conserved Domain Search (https://www.ncbi.nlm.nih.gov/Structure/cdd/wrpsb.cgi, accessed on 29 May 2019) was used to analyze catalytically active sites and conserved domains. SOPMA (https://npsa-prabi.ibcp.fr/cgi-bin/npsa_automat.pl?page=/NPSA/npsa_seccons.html, accessed on 29 May 2019) was used to predict the secondary structure of proteins, SWISS-MODEL (https://swissmodel.expasy.org/interactive, accessed on 29 May 2019) was used to predict protein tertiary structures, and PyMOL 1.1 was used for three-dimensional modeling. A phylogenetic tree was constructed using the neighbor-joining method and bootstrapped 1000 times in MEGA 6.0 (Mega Limited, Auckland, New Zealand).

### 2.4. Expression Profiles of FoCCS1 and FoMnSOD2 at the Different Developmental Stages of F. occidentalis

The expression profiles of *FoCCS1* and *FoMnSOD2* at different developmental stages were determined using the *F. occidentalis* population reared on kidney bean pods. The larval stages (first instar, La-1; second instar on day 1, La2-1; and second instar on day 3, La2-3); pupal stage (Pu; on day 2 after pupation); and adult males and females on days 2 (Fe-2 and Ma-2), 5 (Fe-5 and Ma-5), 8 (Fe-8 and Ma-8), 11 (Fe-11 and Ma-11), and 14 (Fe-14 and Ma-14) after eclosion were collected. Three replicates per instar and 100 *F. occidentalis* individuals per replicate were used. The Primer Design tool was used to design primers for quantitative reverse-transcription PCR (RT-qPCR) of target genes (Appendix A). RT-qPCR was performed on a CFX96TM real-time quantitative PCR system (BioRad, Hercules, CA, USA) using *EF-1a* as an internal reference gene (GenBank: AB277244.1) [28] and FastStart Essential DNA Green Master kit (Roche Diagnostics GmbH, Penzberg, Germany) according to manufacturer’s instructions. The reaction mixture (20 μL) contained 2 μL of cDNA (300 ng/μL), 1 μL each of the upstream and downstream primers, 6 μL of H_2_O, and 10 μL of DNA Green Master Mix. The following PCR program was used: 40 cycles of pre-denaturation at 95 °C for 10 min, denaturation at 95 °C for 30 s, annealing (Tm = 57 °C) for 30 s, and final extension at 72 °C for 30 s.

### 2.5. Effects of Host Shifting on the Expression Levels of FoCCS1 and FoMnSOD2 in F. occidentalis

*F. occidentalis* were reared after host shifting, and samples were collected as described by Liu et al. [25]. Adult females from the experimental *F. occidentalis* population were released into rearing boxes containing kidney bean and broad bean plants, respectively, 3 days after eclosion. There were 12 pots of kidney bean plants per rearing box, and approximately 100 female adult *F. occidentalis* were released per pot of kidney bean. After 24 h of egg laying, all *F. occidentalis* adults were removed. The developmental status of *F. occidentalis* on the host plant was recorded daily. When the hatched larvae developed into second-instar larvae, samples of these second-instar larvae of the F_1_ generation were collected. The remaining larvae continued to be reared to the adult stage. During this period, kidney bean and broad bean plants were replaced every 2 days to ensure continued feeding by *F. occidentalis* larvae. F_1_ generation adult females were collected 3 days after eclosion. The second-instar larvae and adult females reared on kidney bean plants were denoted as PLaF1 and PFeF1, respectively. The second-instar larvae and adult females reared on broad bean plants were denoted as VLaF1 and VFeF1, respectively. Samples from F_2_ and F_3_ generations were collected using the same rearing method. The F_2_ generation samples were denoted as PLaF2, PFeF2, VLaF2, and VFeF2, respectively, and the F_3_ generation samples were denoted as PLaF3, PFeF3, VLaF3, and VFeF3, respectively. In this experiment, the *F. occidentalis* populations reared on kidney bean pods were used as controls, and the corresponding samples were denoted as LaCK and FeCK, respectively. There were three biological replicates per treatment with 100 *F. occidentalis* individuals per replicate. The samples were flash-frozen in liquid nitrogen and stored at −80 °C. Total RNA extraction, cDNA synthesis, and RT-qPCR were performed as described above.

### 2.6. Synthesis of FoCCS1 Double-Stranded RNA

The *FoCCS1* CDS was used (https://www.flyrnai.org/cgi-bin/RNAi_find_primers.pl, accessed on 9 December 2020) to design primers for double-stranded (ds) RNA synthesis. The T7 promoter sequence (TAATACGACTCACTATAGGG), ds*FoCCS1*-F, and ds*FoC**CS1*-R were added to the 5′ ends of the upstream and downstream primers, respectively. The TranscriptAid T7 High Yield Transcription Kit (Thermo Scientific) was used to synthesize *FoC**CS1* dsRNA, and the GeneJET RNA Purification Kit (Thermo Scientific) was used to purify the synthesized dsRNA, which was stored at −80 °C. *eGFP* was used as a negative control (Appendix A).

### 2.7. FoCCS1 RNA Interference and Determination of SOD Enzyme Activity

The F_2_ generation adult females of *F. occidentalis* reared on kidney bean and broad bean plants were used as experimental insects on day 3 after emergence. According to the method described by Zhang et al. [29], *F. occidentalis* were fed 300 µL of dsRNA-free honey solution, 300 ng/µL of ds*eGFP* honey solution, and 300 ng/µL of ds*FoCCS1* honey solution, respectively. The *F. occidentalis* reared on kidney bean plants and broad bean plants feeding on dsRNA-free honey solution were denoted as HK and HB, respectively. Three biological replicates per treatment and 40 adult females per replicate were used. The samples were collected after 24 h of feeding with the dsRNA solution, and the expression level of *FoC**CS1* after RNA interference was determined using RT-qPCR. The F_2_ generation *F. occidentalis* adult females that were transferred to kidney bean plants for rearing were subjected to RNAi, and samples were collected to determine the SOD activity using SOD activity assay kit (Suzhou Keming Biotechnology Co., Ltd., Suzhou, China).

### 2.8. Effects of RNAi on the Survival Rate of Adult Females and the Number of Offspring Nymphs of F. occidentalis

Using the RNAi treatment method described in Section 2.7, adult females were fed ds*eGFP* and ds*FoC**CS1* solutions for 24 h, respectively. Then, 35 RNAi-treated *F. occidentalis* adult females were released onto the kidney bean and broad bean leaves, respectively, to continue rearing. The rearing was performed in a transparent cylindrical plastic cup with a lid (5.5 cm in diameter and 4.0 cm in height, covered with a square hole with a length of 3.5 cm, and sealed with a 200-mesh gauze to ensure air permeability and prevent the thrips from escaping). Kidney bean and broad bean leaves were kept moist by wrapping the petioles with moist absorbent cotton. The survival rate was recorded every 2 days, and the surviving *F. occidentalis* were transferred to fresh kidney bean and broad bean leaves to continue rearing. The survival rates of *F. occidentalis* released back to the host plants and reared for 2, 4, 6, 8, and 10 days were recorded. The bean and broad bean leaves on which adult females had laid eggs were maintained in the artificial climate box, and the number of hatched nymphs was counted after 6 days. The number of larvae hatched on the replaced leaves on days 2, 4, 6, 8, and 10 were also recorded.

### 2.9. Statistical Analyses

Microsoft Excel 2019 was used for data analysis, and relative gene expression was calculated using the 2^−ΔΔCt^ quantitative method [30]. SPSS v.21.0 (IBM, Armonk, NY, USA) was used for one-way analysis of variance, and Tukey’s method was used for multiple comparisons (α = 0.05). The survival rate of adult females and the number of offspring larvae of *F. occidentalis* fed with ds*FoC**CS1* and ds*eGFP* was calculated by independent samples *t*-test. GraphPad Prism v.8.0 (GraphPad Software, San Diego, CA, USA) was used to plot the survival curve of *F. occidentalis* after RNAi treatment. SigmaPlot v.14.0 (Systat Software, SigmaPlot for Windows, San Jose, CA, USA) was used to construct graphs.

## 3. Results

### 3.1. Sequence Characterization of FoCCS1 and FoMnSOD2

Two full-length CDS sequences of the SOD gene of *F. occidentalis* were cloned and identified as belonging to the CCS and MnSOD (SOD2) subfamily, respectively. They were labeled as FoCCS1 (GenBank accession number: MT571493) and FoMnSOD2 (GenBank accession number: MT460164), respectively. The bioinformatics features of FoCCS1 and FoMnSOD2 sequences were markedly different (Table 1). FoCCS1 contains a Cu/ZnSOD signature 2 sequence (GNSGhRlACgiI) and a CCS-specific conserved domain PLN02957, which is described as copper, zinc superoxide dismutase (Table 1 and Figure 1). FoMnSOD2 contains a manganese and iron SOD signature sequence (DvWEHAYY) and a specific conserved domain SodA (Table 1 and Figure 1). Secondary structure prediction showed that α-helices, extended strands, and random coils accounted for 0.66%, 36.30%, and 63.04%, respectively, in FoCCS1 and 56.44%, 3.11%, and 40.44%, respectively, in FoMnSOD2 (Appendix A). The results of protein tertiary structure prediction showed that FoCCS1 had 60.67% homology with the human copper chaperone for SOD1 and had a global model quality estimation (GMQE) value of 0.40. The homology between FoMnSOD2 and *Caenorhabditis elegans* mitochondrial MnSOD2 was 61.54%, and the GMQE value was 0.78. The PyMOL modeling results of FoMnSOD2 protein tertiary structures are shown in Appendix A.

### 3.2. Homologous Alignment and Phylogenetic Analysis

The amino acid sequences of FoCCS1 and FoMnSOD2 were used as query sequences for BLASTp alignment analysis. FoCCS1 showed the highest sequence homology (76%) with CCS of *Thrips palmi* (Thysanoptera; GenBank accession number: XP_034249013.1). It also showed high sequence homology at 60%, 60%, and 59.47%, respectively, with CCS of insects belonging to other orders, such as *Nilaparvata lugens* (GenBank accession number: AQW43017.1), *Cimex lectularius* (GenBank accession number: XP_014251592.1), and *Diprion similis* (GenBank accession number: XP_046741472.1). FoMnSOD2 showed high sequence homology of 96%, 90%, and 69%, respectively, with MnSOD of *F. cephalica* (GenBank accession number: QBH73414.1), *T. palmi* (GenBank accession number: XP_034256995.1), and *Franklinothrips vespiformis* (GenBank accession number: QBH73415.1). It also showed high sequence homology at 64%, 61%, and 67%, respectively, with MnSOD of insects belonging to other orders, such as *Oxya chinensis* (GenBank accession number: QFZ96018.1), *Anoplophora glabripennis* (GenBank accession number: XP_018577318.1), and *Heortia vitessoides* (GenBank accession number: AWX63642.1).

The amino acid sequences of SOD genes belonging to the CCS and MnSOD subfamilies in other insects were downloaded from the NCBI database, and a phylogenetic tree was constructed using the neighbor-joining method in MEGA 6.0. The results showed that FoCCS1 and FoMnSOD2 belonged to two branches of CCS and MnSOD. FoCCS1 is most closely related to *T. palmi* CCS, and with the CCS of *Laodelphax striatellus*, *Apolygus lucorum*, and *Nilaparvata lugens,* and other insects grouped together. FoMnSOD2 is also most closely related to MnSOD of *T. palmi* and *F. vespiformis* (Thysanoptera), and with the MnSOD of *Dastarcus helophoroides*, *Mayetiola destructor*, and *Bombyx mori* and other insects grouped together (Figure 2).

### 3.3. Analysis of Expression Profiles in the Different Developmental Stages

RT-qPCR assay at different developmental stages showed that *FoCCS1* and *FoMnSOD2* were expressed in the larval, pupal, and adult stages. The expression levels of the two genes were higher in adult females than in the larvae, pupae, and adult males (Figure 3). The expression levels of *FoCCS1* in adult females on days 2 (Fe-2), 5 (Fe-5), 8 (Fe-8), 11 (Fe-11), and 14 (Fe-14) were 1.74, 1.73, 3.0, 3.5, and 3.2 times higher than those in the first-instar larvae (La-1), respectively. There was no significant difference in terms of *FoCCS1* expression among the larvae, pupae, and adult males. The expression levels of *FoMnSOD2* in adult females on days 2 (Fe-2), 5 (Fe-5), 8 (Fe-8), 11 (Fe-11), and 14 (Fe-14) were 2.2, 2.6, 3.8, 4.8, and 3.9 times higher than those in the first-instar larvae (La-1), respectively. The expression levels of *FoMnSOD2* in adult males on days 2 (Ma-2) and 14 (Ma-14) were 2.2 and 2.5 times higher than those of La-1 on the, respectively, and the expression levels were similar among other stages.

### 3.4. Expression of FoCCS1 in F. occidentalis Feeding on Different Hosts

The expression levels of *FoCCS1* in *F. occidentalis* larvae feeding on kidney bean and broad bean plants differed after host shifting. The expression level of *FoCCS1* significantly increased in adult females in the F_2_ generation (Figure 4). The expressions of *FoCCS1* in the second-instar larvae of F_2_ and F_3_ generations fed on kidney bean plants were significantly upregulated (3.42-fold and 1.95-fold, respectively) compared with the control (LaCK). However, the expression of *FoCCS1* in the second-instar larvae shifted to broad bean plants for rearing was significantly upregulated only in the F_1_ generation. The expression levels of *FoCCS1* in F_1_ and F_3_ generation adult females shifted to the two host plants for rearing were not significantly different from those in the control. However, the expression levels of *FoCCS1* in F_2_ adult females feeding on kidney bean and broad bean plants were significantly upregulated (2.3-fold and 2.5-fold, respectively).

### 3.5. Expression of FoMnSOD2 in F. occidentalis Feeding on Different Hosts

RT-qPCR revealed that the expression levels of *FoMnSOD2* in *F. occidentalis* were different between those shifted to kidney bean and broad bean plants for rearing (Figure 5). The expression levels of *FoMnSOD2* in the second-instar larvae of the F_1_, F_2_, and F_3_ generations of *F. occidentalis* reared on the kidney bean and broad bean plants were significantly lower than those in the control (LaCK). The expression levels of *FoMnSOD2* in F_1_, F_2_, and F_3_ generation adult females shifted to kidney bean plants were significantly upregulated by 1.9-, 2.2-, and 1.5-fold, respectively, compared with the control (FeCK). However, the expression levels of *FoMnSOD2* in adult females shifted to broad bean plants for rearing did not significantly differ from those in the control.

### 3.6. FoCCS1 Expression Level and SOD Activity Downregulated by RNAi

The results of gene expression and enzyme activity assay indicated that feeding a ds*FoCCS1* solution significantly downregulated the expression level of *FoC**CS1* and SOD activity in adult females of *F. occidentalis* (Figure 6). When *F. occidentalis* shifted to kidney bean plants were treated with the dsRNA solution, the mRNA expression levels of *FoC**CS1* was significantly downregulated by 56.64% and 62.54%, respectively, compared with those in those fed with ds*eGFP* and HK. After RNAi, the mRNA expression of *FoC**CS1* in *F. occidentalis* reared on broad bean plants was significantly downregulated by 50.73% and 36.93%, respectively. The SOD activity in adult *F. occidentalis* females in the ds*FoC**CS1* treatment group was 49.93% and 54.15% lower, respectively, than that in the HK and ds*eGFP* treatment groups.

### 3.7. RNAi Reduced the Survival Rate and Fecundity of Adult F. occidentalis Females

After the mRNA expression of *FoCCS1* was downregulated by RNAi, the survival rate of adult females provided with kidney bean and broad bean leaves for rearing significantly decreased with rearing time (Figure 7A,B). After feeding with ds*FoCCS1* solution, the survival rates of adult females fed on the leaves of kidney bean significantly decreased by 7.86%, 18.52%, 37.60%, and 43.59%, respectively, on days 4, 6, 8, and 10 compared with the ds*eGFP*-treated group. After RNAi treatment, the survival rates of adult females fed on broad bean leaves on days 2, 6, 8, and 10 were 4.32%, 23.02%, 40.52%, and 50.00% lower than those in the ds*eGFP*-treated group, respectively.

With the gradual decrease in the survival rate, the number of offspring larvae hatched on the kidney bean and broad bean leaves showed a decreasing trend (Figure 7C,D). On days 6, 8, and 10 after RNAi treatment, the number of offspring larvae that continued to be fed with kidney bean leaves significantly reduced by 45.48%, 55.90%, and 61.01%, respectively, compared with the control. However, in the treatment group fed with broad bean leaves, the number of offspring larvae decreased significantly by 46.50% and 73.68% on days 8 and 10, respectively. The results indicate that the downregulation of *FoCCS1* mRNA expression by RNAi was not conducive to the feeding adaptation of *F. occidentalis* after host shifting.

## 4. Discussion

Insect SODs include two classes, namely Cn/ZnSOD and MnSOD (SOD2). The molecular conformation of Cn/ZnSOD involves a β-sheet structure, whereas that of MnSOD includes many α-helical structures [31]. CCS is a member of the Cu/ZnSOD family [18]. In the amino acid sequence of *FoCCS1* cloned in this study, α-helical structures accounted for only 0.66%, whereas tertiary structure prediction indicated that this protein mostly comprises β-sheet structures (Appendix A). In the FoMnSOD2 amino acid sequence, α-helical structures accounted for 56.44%. These findings are consistent with the basic structural features of Cn/ZnSOD and MnSOD. The amino acid sequence of CCS1 of *F. occidentalis* is the same as that of CCS of *Coptotermes formosanus* (GenBank: GFG34083.1) [32]. *L. striatellus* (GenBank: RZF38044.1) [33], and *Apolygus lucorum* (GenBank: KAF6204931.1) [34], with a specific Cu/ZnSOD signature 2 (GNSGhRlACgiI) and a specific conserved domain PLN02957 (copper, zinc superoxide dismutase) (Table 1 and Figure 1). Similar to the MnSOD2 of insects, such as *F. vespiformis* [35], FoMnSOD2 contains a specific manganese and iron SOD signature sequence (DvWEHAYY) and a specific conserved domain SodA (Table 1 and Figure 1). Homology alignment and phylogenetic analysis showed that FoCCS1 exhibits high sequence homology with CCS in other insects, such as *T. palmi* (GenBank: XP_034249013.1) and *L. striatellus*. FoMnSOD2 exhibits high sequence homology to MnSODs in other insects, such as *T. palmi* (GenBank: XP_034256995.1), *F. vespiformis* (GenBank: QBH73415.1), and *H. vitessoides* (GenBank: AWX63642.1). Therefore, the cloned FoCCS1 and FoMnSOD2 in this study belong to the CCS (Cu/ZnSOD family) and MnSOD classes, respectively.

Analysis of expression profiles at different developmental stages of *F. occidentalis* showed that the mRNA expression levels of *FoCCS1* and *FoMnSOD2* in adult females were higher than those in adult males, pupae, and larvae. Previous studies have shown that the mRNA levels of *Cu/ZnSOD* and *MnSOD* in *Chironomus riparius* are also higher in adult females than that in eggs, larvae, pupae, and male adults [36]. The expression levels of *Drosophila melanogaster* SOD3 were also higher in females than males [37]. In *F. occidentalis*, the higher mRNA levels of *FoCCS1* and *FoMnSOD2* in adult females than in males may be associated with the significantly longer lifespan of females [38]; the antioxidant systems of long-lived animals exhibit higher SOD activity than shorter-lived animals [39]. In addition, the expression level of Cu/ZnSOD in *Bemisia tabaci* was also higher in the adult stage than in the nymphal and pupal stages [40]. However, the expression level of *Bt-mMnZnSOD* was significantly higher in the fourth instar than in eggs, nymphs, and adults [41]. This may be because SOD is a key enzyme of the antioxidant defense system in insects.

Transcript level analysis of insects after seasonal host shifting showed that antioxidant enzyme genes represented by SOD are actively involved in regulating the anti-adaptation ability of insects to new hosts [8,40,41]. The upregulation of SOD activity and expression of SOD genes effectively helps insects avoid the toxic effects of ROS, such as H_2_O_2,_ from plants, thereby protecting the insects against oxidative damage [13,16,17]. As a member of the Cu/ZnSOD1 family, CCS is a key factor that regulates copper homeostasis and promotes all levels of SOD1 activity in organisms [19,20,21,22]. The results of this study showed that the *FoCCS1* expression levels were significantly upregulated in the F_2_ generation adult females that were shifted to the kidney bean and broad bean plants for rearing. The expression levels of *FoCCS1* were also significantly upregulated in the F_1_ and F_2_ generation larvae that were shifted to the kidney bean plants and the F_1_ generation larvae that were shifted to the broad bean plants. The level of *FoMnSOD2* expression was significantly upregulated only in F_1_, F_2_, and F_3_ generation for adult females that were shifted to kidney bean plants for rearing. Previous studies have shown that SOD plays an active role in regulating the feeding adaptations of *B. tabaci* and aphids after host shifting [42]. For example, the mRNA levels of *Bt-mMnSOD* and *Bt-ecCuZnSOD* were significantly upregulated 24 h after the MEAM1 and Asia II 3 whiteflies were transferred from cotton to tobacco, and the corresponding MnSOD activity and Cu/ZnSOD activity were significantly increased [40]. The activities of Cu/ZnSOD and MnSOD also increased significantly after MEAM1 *B. tabaci* were shifted for feeding from cotton to tobacco for 5 days [41,43]. Transcript level analysis of the aphids *Hyalopterus persikonus* feeding on winter and summer hosts also found that genes, such as SOD, were significantly expressed in aphids feeding on summer hosts [8]. This indicates that SOD plays an important role in regulating the feeding adaptation after insect host shifting. In addition, the present study found that the expression level of *FoMnSOD2* was significantly upregulated only in F_1_, F_2_, and F_3_ generation for adult females that were shifted to kidney bean plants and was significantly downregulated in other treatments, or not different from the control. This may be related to the specific insect state of *F. occidentalis* and the targeted role of the host plant’s antioxidant defense. Only under some special conditions can the MnSOD located on the mitochondria fully exert the function of peroxidase [44].

Decreased SOD activity reportedly reduces antioxidant defense and decreases lifespan in *Drosophila* [45,46]. In addition, inhibiting the mRNA levels and SOD activity of *SOD2* (MnSOD class gene) and *SOD1* (Cu/ZnSOD class gene) by RNAi technology reduced the survival of *Drosophila* [39,47]. In this study, the differences in the mRNA expression levels of *FoCCS1* and *FoMnSOD2* in *F. occidentalis* reared on kidney bean and broad bean plants suggest that *FoCCS1* plays an active role in regulating the feeding adaptation in *F. occidentalis* after host shifting. Therefore, *FoCCS1* mRNA level was further downregulated by RNAi, which showed that SOD activity was also inhibited to a certain extent. This may be because the downregulation of *FoCCS1* mRNA expression reduced the activation of Cu/Zn superoxide dismutase, resulting in the inhibition of SOD1 maturation [19,22]. Moreover, *F. occidentalis* with downregulated *FoCCS1* mRNA levels released in kidney bean and broad bean leaves significantly decreased the survival rate and fecundity of adult females. This finding suggests that the decreased mRNA expression of *FoCCS1* and SOD activity resulted in the inability of *F. occidentalis* to resist a series of plant defense responses related to ROS, which reduced the feeding adaptation ability after host shifting. By interfering with the function of *FoCCS1*, the adaptability of *F. occidentalis* to host shift can decrease. In practice, how to reduce the function of *FoCCS1* will be the focus and direction of future research. The results of this study also provide a reference for the selection of control targets for *F. occidentalis*. Secondly, in the early process of thrips transferring to new hosts is the key time to control, and with the higher control effect, measures should be taken to destroy the adaptability of thrips to the new host as soon as thrips are found in the new host.

## 5. Conclusions

Previous studies on insects, such as aphids and *B. tabaci,* reported that SODs respond positively to a new host, leading to feeding adaptations after host shifting [8,40,41,43]. In the present study, the full-length CDS region of *CCS1* and *MnSOD2* of *F. occidentalis* was cloned and the protein structures were predicted. The expression profiles of *FoC**CS1* and *FoMnSOD2* in different developmental stages of *F. occidentalis* were analyzed. Moreover, the differences in the *FoCCS1* and *FoMnSOD2* mRNA expression levels in *F. occidentalis* that had shifted to the kidney bean and broad bean plants for rearing were examined. RNAi downregulation of *FoCCS1* mRNA expression levels and SOD activity decreased the survival rate and fecundity of *F. occidentalis* adult females. These results indicate that *FoCCS1* plays an important role in regulating the feeding adaptation of *F. occidentalis* after host shifting. These findings provide a reference for future research on indirectly improving insect resistance of crops by reducing the antioxidant capacity of insects.

## Figures and Tables

**Figure 1 insects-13-00782-f001:**
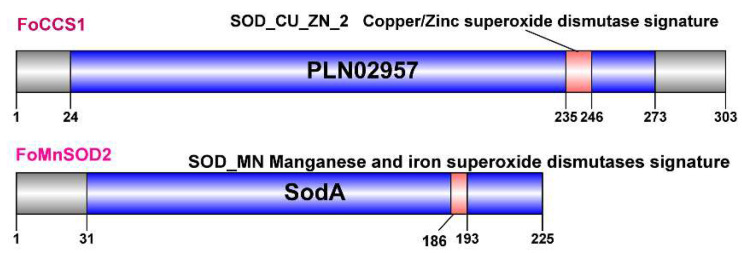
Analysis of the conserved domains of CCS1 and MnSOD2 proteins of *Frankliniella occidentalis* using NCBI Conserved Domain Search (nih.gov). The blue box indicates the superoxide dismutase (SOD) domain, and the red box indicates the signature sequence.

**Figure 2 insects-13-00782-f002:**
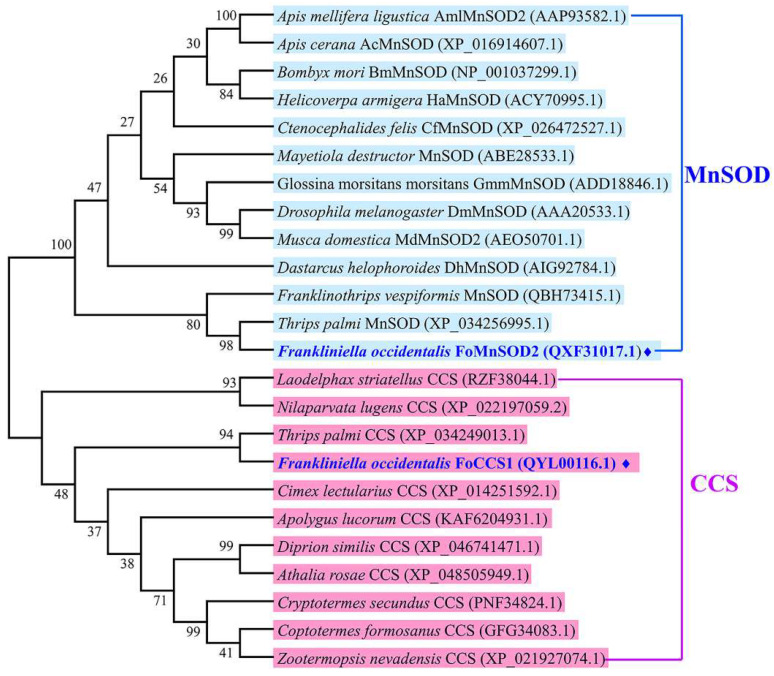
Phylogenetic tree of copper chaperone for superoxide dismutase (CCS) and manganese superoxide dismutase (MnSOD) in *Frankliniella occidentalis* and other insect species using the neighbor-joining method with a bootstrap value of 1000. The branch name of the phylogenetic tree is the species name and GenBank accession number corresponding to each subfamily gene. ♦ represents the amino acid sequence of the gene cloned in this study.

**Figure 3 insects-13-00782-f003:**
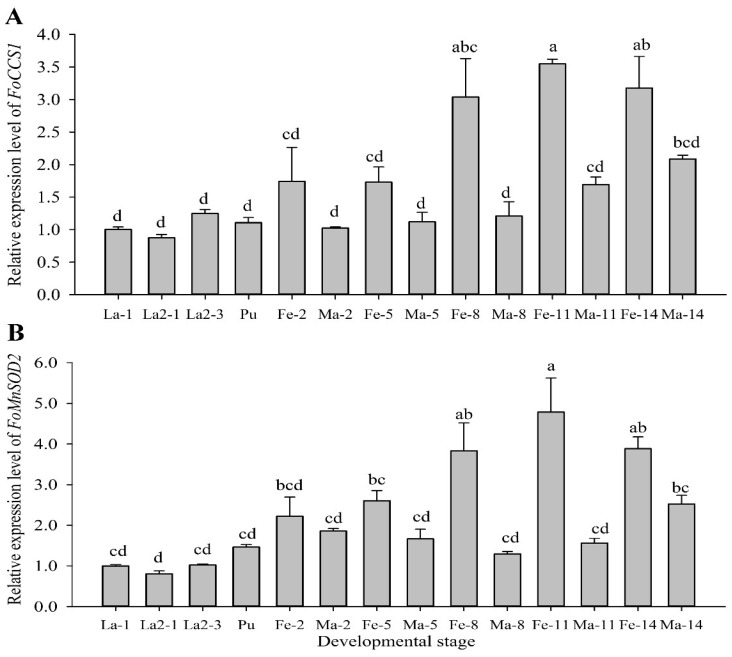
Expression profiles of *Frankliniella occidentalis FoCCS1* and *MnSOD2* genes at different developmental stages. Data are presented as mean ± standard error of the mean (*n* = 3). Significant differences in comparison with the levels in first-instar larvae are indicated by different letters (Tukey’s test, *p* < 0.05).

**Figure 4 insects-13-00782-f004:**
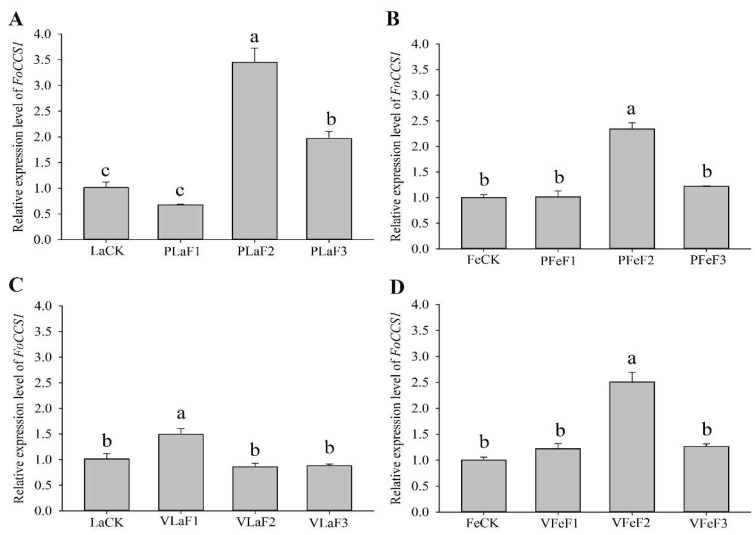
Expression of *FoCCS1* in *Frankliniella occidentalis* feeding on kidney bean and broad bean plants. (**A**), second-instar larvae feeding on kidney bean plants; (**B**), adult females feeding on kidney bean plants; (**C**), second-instar larvae feeding on broad bean plants; (**D**), adult females feeding on broad bean plants. LaCK, second-instar larvae reared on kidney bean pods; FeCk, adult females reared on kidney bean pods; PLaF1–PLaF3, second larvae of F_1_–F_3_ generation reared on kidney bean plants; PFeF1–PfeF3, adult females of F_1_–F_3_ generation reared on kidney bean plants; VLaF1–VlaF3, second larvae of F_1_–F_3_ generation reared on broad bean plants; VFeF1–VFeF3, adult females of F_1_–F_3_ generation reared on broad bean plants. Data are expressed as mean ± standard error of the mean (*n* = 3). Different lowercase letters indicate significant difference among all treatments (Tukey’s test, *p* < 0.05).

**Figure 5 insects-13-00782-f005:**
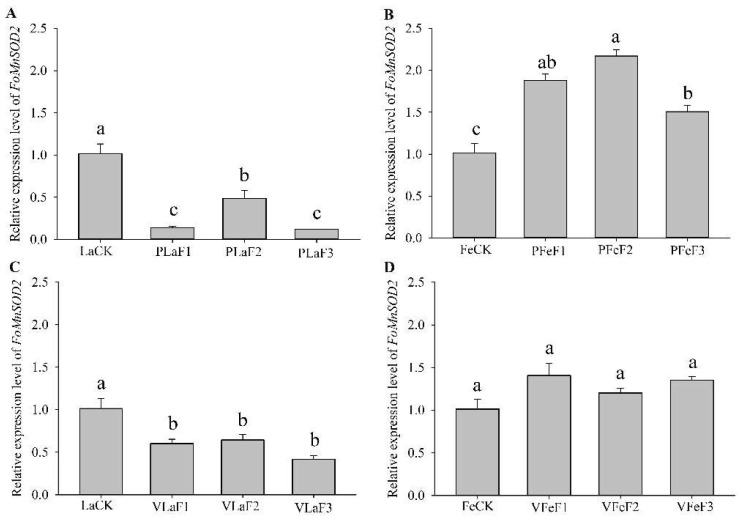
Expression of *FoMnSOD2* in *Frankliniella occidentalis* feeding on kidney bean and broad bean plants. (**A**), second-instar larvae feeding on kidney bean plants; (**B**), adult females feeding on kidney bean plants; (**C**), second-instar larvae feeding on broad bean plants; (**D**), adult females feeding on broad bean plants. LaCK, second-instar larvae reared on kidney bean pods; FeCk, adult females reared on kidney bean pods; PLaF1–PLaF3, second larvae of F_1_–F_3_ generation reared on kidney bean plants; PFeF1–PfeF3, adult females of F_1_–F_3_ generation reared on kidney bean plants; VLaF1–VlaF3, second larvae of F_1_–F_3_ generation reared on broad bean plants; VFeF1–VFeF3, adult females of F_1_–F_3_ generation reared on broad bean plants. Data are expressed as mean ± standard error of the mean (*n* = 3). Different lowercase letters indicate significant difference among all treatments (Tukey’s test, *p* < 0.05).

**Figure 6 insects-13-00782-f006:**
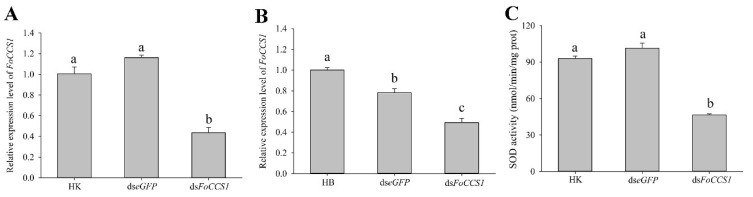
Effects of feeding dsRNA on the *FoCCS1* mRNA expression and SOD activity in *Frankliniella occidentalis*. (**A**,**B**), Efficiency of *FoCCS1* gene silencing in F_2_ adult females of thrips transferred to kidney bean and broad bean plants for feeding, respectively. (**C**), SOD activity of *F*. *occidentalis* reared on kidney bean plants after downregulation of *FoCCS1* expression levels by RNAi. HK and HB: *F*. *occidentalis* reared on kidney bean and broad bean plants and fed with dsRNA-free solution, respectively. Data are expressed as mean ± SE (*n* = 3). Different lowercase letters indicate significant differences among all treatments (Tukey’s test, *p* < 0.05).

**Figure 7 insects-13-00782-f007:**
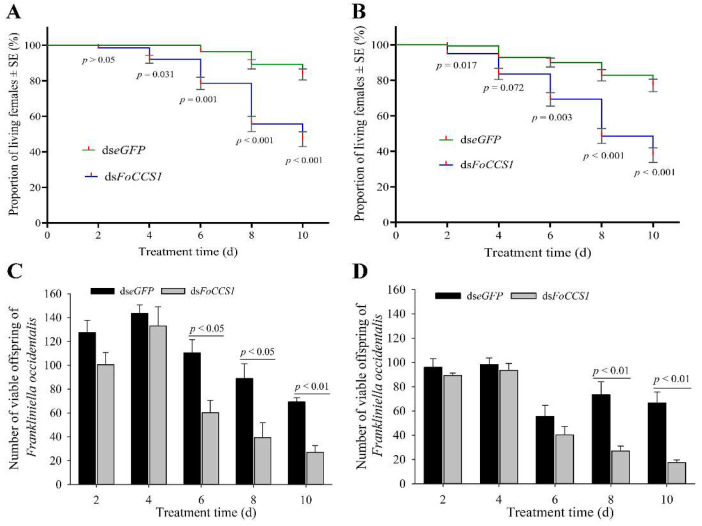
Effects of RNAi downregulation of *FoCCS1* expression on survival and offspring number of the *Frankliniella occidentalis* adult females shifted to feed on kidney bean and broad bean plants. (**A**,**C**), survival curve and number of offspring larvae of *F. occidentalis* reared on kidney bean plants after being fed with dsRNA solution. (**B**,**D**), survival curve and number of offspring larvae of *F. occidentalis* reared on broad bean plants after being fed with dsRNA-solution. Data are presented as the mean ± standard error of the mean (*n* = 4) and compared using the independent sample *t*-test. *p* < 0.05 indicates statistical significance.

**Table 1 insects-13-00782-t001:** Basic biological information for *Frankliniella occidentalis* CCS and MnSOD2 sequences.

Gene	ORF Finder	Number of Amino Acids	Molecular Weight	Theoretical pI	Phosphorylation Site	Signature
*FoCCS1*	912 bp	303	31,978.98	5.84	2 (S), 0 (T), 0 (Y)	GNSGhRlACgiI
*FoMnSOD2*	678 bp	226	25,514.35	8.43	1 (S), 2 (T), 2 (Y)	DvWEHAYY

Note: The data represent the results of the ORF finder tool, ExPASy-ProtParam, PROSITE, and KinasePhos predictive analysis of the nucleotide or protein sequences of the two genes.

## Data Availability

The data that support the findings of this study are available from the corresponding author upon reasonable request.

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
