# Peer review of "Copper Chaperone for Superoxide Dismutase FoCCS1 in Frankliniella occidentalis May Be Associated with Feeding Adaptation after Host Shifting"

_insects, 2022, doi:10.3390/insects13090782_

Round 1

Reviewer 1 Report

The current addressed most of all my concerns, while still two minor comments:

1. Line 108, authors did not descirbed which bean pod was provided to the experimental population.

2. The significance of this study should be discussed as mentioned in comment 3 of last version.

Author Response

Response to Reviewer 1 Comments

Point 1: 1. Line 108, authors did not descirbed which bean pod was provided to the experimental population.

Response 1: We appreciate your reminder. We have modified [Page 3, line 108]. Kidney bean pod.

Point 2: The significance of this study should be discussed as mentioned in comment 3 of last version.

Response 2: We have supplied in the section of discussion.  As follows (or please see the manuscript):

By interfering with the function of FoCCS1, the adaptability of F. occidentalis to host shift can decrease. In the practice, how to reduce the function of FoCCS1 will be the focus and direction of future research. The results of this study also provide a reference for the selection of control targets for F. occidentalis. Secondly, in the early process of thrips transferring to new hosts is the key time to control, and with the higher control effect, measures should be taken to destroy the adaptability of thrips to the new host as soon as thrips found in the new host.

Thank you again

Best regards.

All authors

Reviewer 2 Report

It is well known that CCS works with SOD1. Therefore, it is understandable that the conclusions are not significantly different from the previous versions of this paper. In this paper, MnSOD, which is often referred to as SOD2, is referred to as MnSOD1, but I think it is easier for many readers to rewrite all of them as SOD2. Also, there are some mistakes in the chart. For example, CSS1 on the vertical axis of Fig. 4b is originally CCS1. I think this Fig4 will be the central chart in this paper, but the size of the vertical axis of Fig4a-d is not unified. This also applies to Fig5, Fig6ab, and Fig7c-d.

The result of Tukey's test in Fig. 6A is a and c where it should be represented by a and b.

Author Response

Response to Reviewer 2 Comments

Point 1: It is well known that CCS works with SOD1. Therefore, it is understandable that the conclusions are not significantly different from the previous versions of this paper. In this paper, MnSOD, which is often referred to as SOD2, is referred to as MnSOD1, but I think it is easier for many readers to rewrite all of them as SOD2.

Response 1: Thanks again to the reviewers for their very careful review of the article. Based on your suggestion, we checked the literature again, and indeed as you said, MnSOD present in mitochondria is usually classified and annotated as SOD2. Therefore, FoMnSOD1 was changed to FoMnSOD2 in this paper.

Point 2: Also, there are some mistakes in the chart. For example, CSS1 on the vertical axis of Fig. 4b is originally CCS1. I think this Fig4 will be the central chart in this paper, but the size of the vertical axis of Fig4a-d is not unified. This also applies to Fig5, Fig6ab, and Fig7c-d. The result of Tukey's test in Fig. 6A is a and c where it should be represented by a and b.

Response 2: Based on your comments, we have corrected the problems in Figure 1, Figure 2, Figure 3, Figure 4, Figure 5, Figure 6 and Figure 7. For revised figures, please see the manuscript.

Thank you again

Best regards.

All authors

Round 2

Reviewer 2 Report

I think this revision meets the criteria for publication. The vertical axis of the graph is also correctly corrected.

This manuscript is a resubmission of an earlier submission. The following is a list of the peer review reports and author responses from that submission.

Round 1

Reviewer 1 Report

Insects are known to have much SOD. Therefore, it is of great scientific significance to study and understand insect SOD. However, the authors make a fundamental mistake: SOD1 is a small protein containing around 150 amino acids in most insects. Much information on this insect (Frankliniella occidentalis) is available at NCBI (https://www.ncbi.nlm.nih.gov/genome/?term=Frankliniella%20occidentalis) and elsewhere. Therefore, as a reviewer, I could see much protein data on this insect. As a result, the protein claimed by the authors to be SOD1 does not appear to be SOD1. For example, XP_026285304.1 is a good candidate for SOD1.

There are many proteins with SOD_Cu domains, and there may be other proteins with SOD_Cu domains besides SOD. Biochemical verification of what kind of protein SOD1 is also necessary.

Reviewer 2 Report

The MS entitled with “Superoxide dismutase FoCu/ZnSOD in Frankliniella occidentalis may be associated with feeding adaptation after host shifting” focus on Cu/ZnSOD1 and MnSOD1 of Frankliniella occidentalis on host shifting. It will provide a reference for 18 analyzing the adaptive mechanism of F. occidentalis after host shifting. However, the current version should be revised. The comments are as follows:

1. I am confused about the host shifting, did the author mean that thrips were reared on kidney bean and then were transferred to broad bean? I think the author should make this clear firstly.

2. Two genes, Cu/ZnSOD1 and MnSOD1, were cloned in the first part, while the RNAi experiment was only conducted for Cu/ZnSOD1. Author should explain this.

3. After finishing these experiment, what suggestions would be provided for the control of Frankliniella occidentalis?

4. Although the words “PLaF1, PFeF1, VLaF1 and VFeF1” were described in the methods part, authors should add these description in the figure notes to make it easily read.

5. For Figure 7, statistical analysis method should be added.